# Rotating behind Security: A Lightweight Authentication Protocol Based on IoT-Enabled Cloud Computing Environments

**DOI:** 10.3390/s22103858

**Published:** 2022-05-19

**Authors:** Tsu-Yang Wu, Qian Meng, Saru Kumari, Peng Zhang

**Affiliations:** 1College of Computer Science and Engineering, Shandong University of Science and Technology, Qingdao 266590, China; wutsuyang@gmail.com (T.-Y.W.); MQ15753683129@163.com (Q.M.); 2Department of Mathematics, Chaudhary Charan Singh University, Meerut 250004, India; saryusiirohi@gmail.com

**Keywords:** IoT, cloud computing, authentication protocol, formal security analysis

## Abstract

With the rapid development of technology based on the Internet of Things (IoT), numerous IoT devices are being used on a daily basis. The rise in cloud computing plays a crucial role in solving the resource constraints of IoT devices and in promoting resource sharing, whereby users can access IoT services provided in various environments. However, this complex and open wireless network environment poses security and privacy challenges. Therefore, designing a secure authentication protocol is crucial to protecting user privacy in IoT services. In this paper, a lightweight authentication protocol was designed for IoT-enabled cloud computing environments. A real or random model, and the automatic verification tool ProVerif were used to conduct a formal security analysis. Its security was further proved through an informal analysis. Finally, through security and performance comparisons, our protocol was confirmed to be relatively secure and to display a good performance.

## 1. Introduction

The Internet of things (IoT) [1,2,3,4] is the “Internet connected by all things”. It is the combination of networks and various sensing devices and compose a huge network that can interconnect users and everything whenever and wherever. The emergence of IoT has driven the development of many industries, such as transportation, agriculture, medical treatment, and artificial intelligence [5,6,7]. It has since made significant advancements and can connect various devices with limited resources, and massive amounts of data can be shared through the Internet.

Cloud computing [8,9] can connect a large number of resources, such as computation, software, and storage resources, to compose a large virtually shared resource pool [10]. Its core idea is to continuously lower the processing load of user terminals by increasing the processing capacity of the “cloud”, allowing users to exploit the “cloud’s” strong computing processing capacity on demand. With cloud computing, users can access applications on any device that can connect to the Internet [11]. The progress of cloud computing technology has penetrated all aspects of people’s lives and significantly increased the level of convenience during daily life.

In real life, the resource, computing, and communication capabilities of IoT devices are limited. To address these limitations, cloud computing, as a key technology, provides an efficient platform for effectively analyzing, managing, and storing the data generated by IoT devices. Mobile devices allow users to access the cloud server resources at any time from any location. Figure 1 shows the architecture of IoT-enabled cloud computing. This architecture has three entities: control server, user, and cloud server. The cloud server provides the services requested by users conveyed through user IoT devices. The control server is a trusted organization that authorizes users and the cloud server and creates system parameters during the registration phase. In addition, when users intend to obtain the cloud server service, the control server monitors the authentication process, and with help from the control server, the three parties can consult a session key, which the user uses to obtain and enjoy the service of the cloud server.

### Motivation

In IoT-enabled cloud computing environments [12,13,14,15], information is transmitted to the public channel, which is open and unprotected, and users are vulnerable to attackers when obtaining services, resulting in privacy data disclosure issues. Therefore, when users want to obtain cloud services, they must complete identity authentication and establish a key to protect the information from disclosure and tampering. At present, some scholars also use quick response (QR) codes [16] to solve these problems. Many scholars proposed authentication protocols [13,17,18,19] for this environment. However, these protocols typically have security problems, such as an inability to provide perfect forward security, suffering from man-in-the-middle (MITM) and temporary value disclosure attacks. In addition, the power of IoT devices is limited, and reducing the calculation of such devices is necessary.

In this paper, we designed a lightweight authentication protocol to solve the above problems. Both formal and informal security analyses were conducted to verify the security of our protocol. Through security and performance comparisons, our protocol demonstrated a good performance and satisfied the security requirements in IoT-enabled cloud computing environments.

The remainder of this paper is organzed as follows. Section 2 discusses related work. Section 3 presents our protocol in detail. Section 4 introduces a safety analysis. Section 5 presents some security and performance comparisons, and Section 6 presents our conclusions.

## 2. Related Work

This section reviews authentication and key agreement (AKA) protocols [13,17,18,19,20,21,22,23,24,25,26] applied in IoT, cloud computing, and IoT-enabled cloud computing environments. A summary of existing protocols is shown in Table 1. Turkanovic et al. [22] designed an AKA scheme for IoT environments, which are dedicated to the identity authentication of users in wireless sensor networks. However, Wazid et al. [23] testified that Turkanovic et al.’s scheme [22] was unable to prevent insider and user impersonation attacks. Subsequently, Wu et al. [24] designed an AKA protocol and declared that it could prevent many common attacks. Unfortunately, Sadri et al. [27] indicated that Wu et al.’s protocol was unable to resist sensor capture and denial of service attacks (DoS) and was, thus, unable to provide perfect forward security.

Tsai and Lo [25] designed an anonymous AKA scheme for cloud computing environments. They use bilinear pairing to design the scheme, and without the assistance of a control server, users can directly obtain the services of the distributed cloud server. However, He et al. [28] proved that their scheme cannot resist server impersonation attacks. Irshad et al. [26] designed an AKA scheme using a bilinear pairing method. Unfortunately, Xiong et al. [29] verified that Irshad et al.’s [26] lacked user registration and revocation phases. Xiong et al. [29] designed an enhanced scheme and claimed that it can prevent many common attacks.

Amin et al. [13] also designed a protocol applicable to distributed cloud computing environments. However, Challa et al. [30] testified that their protocol cannot prevent insider and impersonation attacks. Martinez et al. [17] designed a lightweight AKA scheme for cloud computing environments. Unfortunately, Yu et al. [31] determined that the scheme cannot prevent impersonation and session key exposure attacks or achieve mutual authentication. Zhou et al. [18] proposed a lightweight AKA scheme. However, Wang et al. [32] proved that their protocol cannot provide perfect forward security and cannot prevent replay, impersonation, and temporary value disclosure attacks. Kang et al. [19] designed a protocol suitable for IoT-enabled cloud computing environments, which supports the authentication of IoT devices. However, Huang et al. [33] verified that it was unable to resist offline password-guessing attacks.

## 3. The Proposed Protocol

This section introduces our protocol. It includes three phases: (1) user registration, (2) cloud server registration, and (3) login and authentication. The following subsections describe each in detail. Table 2 lists the symbols used in the protocol.

### 3.1. System Model

Our IoT-enabled cloud computing model includes three entities, namely user, cloud server, and control server. The information exchange between each entity is shown in Figure 2. (1)User: The user can use IoT devices to obtain cloud server services. We allow the user to be an untrusted entity, which means that they may be a legitimate user but may obtain services or launch attacks maliciously.(2)Cloud server: The cloud server provides the services requested by users conveyed through user IoT devices. It is a semi-trusted entity, in the sense that it may misbehave on its own but does not conspire with either of the participants.(3)Control server: The control server is responsible for registering users and cloud server, assisting users and cloud server in completing authentication and in establishing a session key in the login and authentication phase. It is a semi-trusted entity, in the sense that it may misbehave on its own but does not conspire with either of the participants.

The purpose of our protocol is to realize mutual authentication and to establish a session key between the user and cloud server with the help of the control server. Figure 2 shows the exchange of information. The specific process is referred to in Section 3.4 (Login and Authentication Phase).

### 3.2. User Registration Phase

At this phase, $U_{i}$ registers with CS as a legal user. The user transmits the parameters calculated by themselves to CS via a secure channel and finally obtains the smart card issued by CS. Figure 3 detail the process. The specific process is as follows:(1)$U_{i}$ chooses $ID_{i}$, $PW_{i}$, and $B_{i}$; calculates $Gen\left(B_{i}\right)=\left(\sigma_{i},\tau_{i}\right)$ and $HPW_{i}=h\left(PW_{i}\|\sigma_{i}\right)$; and then sends $\left\{ID_{i},HPW_{i}\right\}$ to control server CS through a secure channel.(2)CS checks $U_{i}$’s identity. If the identity is new, CS selects a random value $n_{i}$ and computes $TID_{i}=h\left(ID_{i}\right)$, $A_{1}=h\left(ID_{CS}\|HPW_{i}\right)⊕\left(n_{i}⊕K_{j}\right)$, stores $\left\{TID_{i},HPW_{i}\right\}$ in the database, stores $\left\{A_{1},ID_{CS}\right\}$ in smart card SC, and then sends SC to $U_{i}$ through a secure channel.(3)After receiving message $\left\{A_{1},ID_{CS}\right\}$ sent by CS, $U_{i}$ calculates $A_{2}=h\left(ID_{i}\|HPW_{i}\right)$ and then stores $\left\{A_{2},Gen\left(\cdot \right),Rep\left(\cdot \right),\tau_{i}\right\}$ in SC.

### 3.3. Cloud Server Registration Phase

At this phase, cloud server $S_{j}$ needs to register with CS as a legal entity. It sends its own parameters to CS via a secure channel, obtains the parameters calculated by the CS, and stores them in its own memory. Figure 4 shows specific the process. The specific process is as follows: (1)$S_{j}$ selects its identity $SID_{j}$ and random number $n_{j}$ and then sends $\left\{SID_{j},n_{j}\right\}$ to CS through a secure channel.(2)CS checks the identity of $S_{j}$. If $S_{j}$ is unregistered, then CS selects a pseudo identity $QID_{j}$ for $S_{j}$, calculates $A_{3}=h\left(SID_{j}\|K_{j}⊕n_{j}\right)$, and stores $\left\{QID_{j},n_{j}\right\}$ in its memory. Then, CS sends $\left\{QID_{j},A_{3}\right\}$ to $S_{j}$ through a secure channel.(3)$S_{j}$ calculates $A_{3}^{*}=A_{3}⊕SID_{j}$ and stores $\left\{A_{3}^{*},QID_{j}\right\}$ in its memory.

### 3.4. Login and Authentication Phase

At this phase, the control server CS verifies the identity of the user $U_{i}$ and cloud server $S_{j}$. After verification, the three establish a common session key for future communication. The specific process is shown in the Figure 5. The specific process is as follows:(1)$U_{i}$ inputs $ID_{i}$ and $PW_{i}$; imprints $B_{i}$; computes $Rep\left(B_{i},\tau_{i}\right)=\sigma_{i}$, $HPW_{i}=h\left(PW_{i}\|\tau_{i}\right)$, $A_{2}^{\prime }=h\left(ID_{i}\|HPW_{i}\right)$; and checks the legitimacy of $U_{i}$’s identity by verifying $A_{2}^{\prime }\overset{?}{=}A_{2}$. If this is valid, $U_{i}$ then chooses a random value $r_{i}$ and timestamp $TS_{1}$ and computes $\left(n_{i}⊕x\right)=A_{1}⊕h\left(ID_{CS}\|HPW_{i}\right)$, $B_{1}=r_{i}⊕h\left(ID_{CS}\|HPW_{i}⊕SID_{j}\right)$, $B_{2}=SID_{j}⊕h\left(ID_{CS}\|HPW_{i}\right)$, and $B_{3}=h(TID_{i}\|ID_{CS}\left\|n_{i}⊕x\right)⊕HPW_{i}$. Subsequently, $M_{1}=\left\{TID_{i},A_{1},B_{1},B_{2},B_{3},TS_{1}\right\}$ is sent to $S_{j}$ through an open channel.(2)After receiving $U_{i}$’s message, $S_{j}$ checks timestamp $|TS_{1}-TS_{c}|≦\Delta T$. If the timestamp is valid, $S_{j}$ then selects a random number $r_{j}$ and timestamp $TS_{2}$. $S_{j}$ calculates $A_{3}=SID_{j}⊕A_{3}^{*}$, $B_{4}=r_{j}⊕h\left(A_{3}\|SID_{j}\right)$, and $B_{5}=h(r_{j}\|A_{3}\left\|SID_{j}\right)$ and then sends message $M_{2}=\left\{M_{1},QID_{j},B_{4},B_{5},TS_{2}\right\}$ to CS through an open channel.(3)After receiving $M_{2}$, CS checks timestamp $|TS_{2}-TS_{c}|≦\Delta T$. If the verification passes, CS finds $HPW_{i}$ according to $TID_{i}$; computes $SID_{j}=B_{2}⊕h\left(ID_{CS}\|HPW_{i}\right)$, $r_{i}=B_{1}⊕h\left(ID_{CS}\|HPW_{i}⊕SID_{j}\right)$, and $B_{3}^{\prime }=h(TID_{i}\|ID_{CS}\left\|n_{i}⊕x\right)⊕HPW_{i}$; and verifies $U_{i}$’s identity by checking $B_{3}^{\prime }\overset{?}{=}B_{3}$. If valid, CS then indexes $n_{j}$ according to the value of $QID_{j}$; computes $A_{3}=h\left(SID_{j}\|x⊕n_{j}\right)$, $r_{j}=B_{4}⊕h\left(A_{3}\|SID_{j}\right)$, and $B_{5}^{\prime }=h(r_{j}\|A_{3}\left\|SID_{j}\right)$; and checks $B_{5}^{\prime }\overset{?}{=}B_{5}$. If valid, CS then selects $r_{k},TS_{3}$ computes $SK=h(r_{i}⊕HPW_{i}\|r_{j}\|r_{k}\|SID_{j})$, $B_{6}=\left(r_{i}⊕HPW_{i}\right)⊕A_{3}$, $B_{7}=h(A_{3}\|r_{j}\left\|SID_{j}\right)⊕r_{k}$, $B_{8}=h(r_{j}\|r_{k}\|SK\|TS_{3})$, $\left(n_{i}⊕x\right)=A_{1}⊕h\left(ID_{CS}\|HPW_{i}\right)$, $B_{9}=h\left(n_{i}⊕x\|SID_{j}\right)⊕r_{j}$, and $B_{10}=h\left(HPW_{i}\|r_{i}\right)⊕r_{k}$, $B_{11}=h(SK\|n_{i}⊕x\|r_{k}\left\|r_{j}\right)$ and sends message $M_{3}=\left\{B_{6},B_{7},B_{8},B_{9},B_{10},B_{11},TS_{3}\right\}$ to $S_{j}$ through an open channel.(4)After receiving $M_{3}$, the cloud server checks the timestamp $|TS_{3}-TS_{c}|≦\Delta T$. If the timestamp is valid, $S_{j}$ then computes $\left(r_{i}⊕HPW_{i}\right)=B_{6}⊕A_{3}$, $SK=h(r_{i}⊕HPW_{i}\|r_{j}\|r_{k}\|SID_{j})$, and $B_{8}^{\prime }=h(r_{j}\|r_{k}\|SK\|TS_{3})$, and checks $B_{8}\overset{?}{=}B_{8}$. If true, $S_{j}$ sends message $M_{4}=\left\{B_{9},B_{10},TS_{4}\right\}$ to $U_{i}$ through an open channel.(5)$U_{i}$ checks timestamp $|TS_{4}-TS_{c}|≦\Delta T$. If the verification passes, $U_{i}$ then computes $r_{j}=h\left(n_{i}⊕x\|SID_{j}\right)⊕B_{9}$, $r_{k}=h\left(HPW_{i}\|r_{i}\right)⊕B_{10}$, $SK=h(r_{i}⊕HPW_{i}\|r_{j}\|r_{k}\|SID_{j})$, and $B_{11}^{\prime }=h(SK\|n_{i}⊕x\|r_{k}\left\|r_{j}\right)$ and checks $B_{11}^{\prime }\overset{?}{=}B_{11}$. If the verification passes, $U_{i}$ then computes $B_{12}=h\left(SK\|r_{j}\right)$ and sends $M_{5}=\left\{B_{12}\right\}$ to $S_{j}$.(6)$S_{j}$ computes $B_{12}^{\prime }=h\left(SK\|r_{j}\right)$ and checks $B_{12}^{\prime }\overset{?}{=}B_{12}$. If the verification passes, then $S_{j}$ stores SK for future communication.

## 4. Security Analysis

This section presents an informal security analysis and describes a formal analysis using ProVerif and the real or random (ROR) model. The subsections introduce these topics.

### 4.1. Attacker Model

We define the attacker’s ability based on the C-K model [35], which is an extension of the D-Y model [36]. The following features of an attacker A are defined:(1)A is assumed to be capable of blocking, modifying, and eavesdropping on messages transmitted on the open channel. It has complete control over communications between the various participants.(2)A can be a malicious insider on the control server and can obtain the content stored in the control server by the user or cloud server.(3)A can disclose the established session key, long-term key, and session state.(4)A can guess the user’s password or identity, but A is unable to guess the user’s identity or password simultaneously in polynomial time.(5)A may extract the information of a user’s SC using power analysis.

### 4.2. Formal Security Analysis

We use the ROR model and the automated verification tool ProVerif to conduct a formal security analysis to testify that the protocol is secure and correct.

#### 4.2.1. ROR Model

The protocol security is demonstrated using the ROR model [4,37]. The security is verified by calculating the probability of session key SK.

The protocol comprises three parties: user, cloud server, and control server. In this model, $\Pi_{U_{i}}^{x}$, $\Pi_{s_{j}}^{y}$, and $\Pi_{CS}^{z}$ are the *x*th user, *y*th cloud server, and *z*th control server, respectively. Suppose attacker A’s query capabilities include the following: $Z=\Pi_{U_{i}}^{x}$, $\Pi_{S_{j}}^{y}$, and $\Pi_{CS}^{z}$.

Execute(Z): Assuming an attacker A executes the query, they can capture messages on the open channel.

Send(Z,M): Assuming an attacker A executes the query, they transfer *M* to *Z* and receive an answer from *Z*.

Hash(string): Suppose an attacker A executes the query; they enter a string and obtain a hash value.

Corrupt(Z): Assuming an attacker A executes the query, they obtain the private value of an entity, for example, a long-term key and temporary information of the user’s SC.

Test(Z): Assume that an attacker A executes the query and tosses a coin *C* into the air. If *C* equals 1, A obtains SK. Otherwise, A obtains a string.

**Theorem** **1.**
*If A executes queries Execute(Z), Send(Z,M), Hash(string), Corrupt(Z), and Test(Z), the probability P of A cracking the protocol is $Adv_{A}^{P}\left(\xi \right)\leq q_{send}/2^{l-2}+3q_{hash}^{2}/2^{l-1}+2max\left\{c^{\prime }\cdot q_{send}^{s^{\prime }},q_{send}/2^{l}\right\}$. Here, $q_{send}$ refers to the numbers of times the queries executed, $q_{hash}$ is the execution time of the hash function, l is the bit length of biological information [38], and $c^{\prime }$ and $s^{\prime }$ are two constants.*


**Proof.** The ROR model played $GM_{0},GM_{1},GM_{2},GM_{3},GM_{4}$. $Succ_{A}^{GM_{i}}\left(\xi \right)$ is the probability that A can win $GM_{0}$–$GM_{4}$. The following are the specific query steps in the game: $GM_{0}$: $GM_{0}$ represents the first round of the game, which starts by flipping *C*. $GM_{0}$ cannot execute any queries; hence, the probability that A can break *P* is as follows:
(1)$$Adv_{A}^{P}\left(\xi \right)=\left|2Pr\left[Succ_{A}^{GM_{0}}\left(\xi \right)\right]-1\right|.$$$GM_{1}$: $GM_{1}$ is for the $GM_{0}$-added Execute(Z) operation, and A can be used only when $GM_{1}$ intercepts the messages $M_{1}$–$M_{5}$ transmitted over the open channel. Then, because the values of $HPW_{i},r_{i},r_{j},r_{k}$ and $SID_{j}$ cannot be obtained, A cannot obtain the session key through the Test(Z) query. Thus, $GM_{1}$’s probability is the same as $GM_{0}$.
(2)$$Pr\left[Succ_{A}^{GM_{1}}\left(\xi \right)\right]=Pr\left[Succ_{A}^{GM_{0}}\left(\xi \right)\right].$$$GM_{2}$: $GM_{2}$ extends $GM_{1}$ by adding the Send(Z,M) query. The probability of $GM_{2}$ is calculated using Zipf’s law [39].
(3)$$|Pr\left[Succ_{A}^{GM_{2}}\left(\xi \right)\right]-Pr\left[Succ_{A}^{GM_{1}}\left(\xi \right)\right]|\leq q_{send}/2^{l}.$$$GM_{3}$: $GM_{3}$ is for the $GM_{2}$-added Hash(string) operation and deleted Send(Z,M) operation. $GM_{3}$’s probability can be obtained using the birthday paradox.
(4)$$|Pr\left[Succ_{A}^{GM_{3}}\left(\xi \right)\right]-Pr\left[Succ_{A}^{GM_{2}}\left(\xi \right)\right]|\leq q_{hash}^{2}/2^{l+1}.$$$GM_{4}$: In $GM_{4}$, a security analysis on two events is conducted to testify the security of the session key. (1) A obtains CS’s long-term key *x*; (2) A obtains the temporary information. This demonstrates that our protocol can guarantee perfect forward security and prevent temporary information disclosure attacks.
(1)Perfect forward security: A with $\Pi_{CS}^{Z}$ to obtain *x* of CS or use $\Pi_{U_{i}}^{x}$, $\Pi_{S_{j}}^{y}$ to obtain private values.(2)Temporary information disclosure attack: A utilizes $\Pi_{U_{i}}^{x}$, $\Pi_{S_{j}}^{y}$ or $\Pi_{CS}^{Z}$ to obtain the random number of three entities.For the first case, even if A obtains *x* or some private values, they cannot calculate $HPW_{i},r_{i},r_{j},r_{k}$, or $SID_{j}$. Therefore, A cannot calculate SK, where $SK=h(r_{i}⊕HPW_{i}\|r_{j}\|r_{k}\|SID_{j})$. For the second case, even if A obtains $r_{i}$ but $HPW_{i},r_{j},r_{k}$ and $SID_{j}$ are private, SK is incalculable. Similarly, even if A can obtain $r_{j}$ or $r_{k}$, SK is also incalculable. Thus, the probability of $GM_{4}$ is obtained:
(5)$$|Pr\left[Succ_{A}^{GM_{4}}\left(\xi \right)\right]-Pr\left[Succ_{A}^{GM_{3}}\left(\xi \right)\right]|\leq q_{send}/2^{l}+q_{hash}^{2}/2^{l+1}.$$$GM_{5}$: In $GM_{5}$, A queries the parameters $\left\{A_{1},A_{2},ID_{CS},Gen\left(\cdot \right),Rep\left(\cdot \right),\tau_{i}\right\}$ in the smart card by executing Corrupt(Z). This proves that our protocol can protect against offline password-guessing attacks. A attempts to guess $A_{2}=h\left(ID_{i}\|HPW_{i}\right)$, where $HPW_{i}=h\left(PW_{i}\|\tau_{i}\right)$. However, $ID_{i}$ and $HPW_{i}$ are private. The probability that A can guesses *l* bit of biological information is $1/2^{l}$. From Zipf’s law [39], when $q_{send}\leq 106$, the probability that A can guess the password is more than 1/2. Thus, the probability of $GM_{5}$ can be obtained:
(6)$$|Pr\left[Succ_{A}^{GM_{5}}\left(\xi \right)\right]-Pr\left[Succ_{A}^{GM_{4}}\left(\xi \right)\right]|\leq max\left\{C^{\prime }\cdot q_{send}^{s^{\prime }},q_{send}/2^{l}\right\}$$$GM_{6}$: $GM_{6}$ confirms that the protocol can prevent impersonation attacks. A queries $h(r_{i}⊕HPW_{i}\|r_{j}\|r_{k}\|SID_{j})$, and the game ends. Therefore, the probability of $GM_{6}$ can be obtained:
(7)$$|Pr\left[Succ_{A}^{GM_{6}}\left(\xi \right)\right]-Pr\left[Succ_{A}^{GM_{5}}\left(\xi \right)\right]|\leq q_{hash}^{2}/2^{l+1}.$$Because A’s probability of success is the same as that of failure (i.e., (1)–(2)), A’s probability of obtaining the session key is
(8)$$Pr[Succ_{A}^{GM_{6}}\left(\xi \right)]=1/2.$$From all these formulas,
(9)$$\begin{array}{cc} 1/2Adv_{A}^{P}\left(\xi \right)\; & =|Pr[Succ_{A}^{GM_{0}}\left(\xi \right)]-1/2|\\ \; & =|Pr\left[Succ_{A}^{GM_{0}}\left(\xi \right)\right]-Pr\left[Succ_{A}^{GM_{6}}\left(\xi \right)\right]|\\ \; & =|Pr\left[Succ_{A}^{GM_{1}}\left(\xi \right)\right]-Pr\left[Succ_{A}^{GM_{6}}\left(\xi \right)\right]|\\ \; & \leq \sum_{i=0}^{5}\left|Pr\left[Succ_{A}^{GM_{i+1}}\left(\xi \right)\right]-Pr\left[Succ_{A}^{GM_{i}}\left(\xi \right)\right]\right|\\ \; & =q_{send}/2^{l-1}+3q_{hash}^{2}/2^{l}+max\left\{c^{\prime }\cdot q_{send}^{s^{\prime }},q_{send}/2^{l}\right\} \end{array}$$Consequently, we obtain
(10)$$Adv_{A}^{P}\left(\xi \right)\leq q_{send}/2^{l-2}+3q_{hash}^{2}/2^{l-1}+2max\left\{c^{\prime }\cdot q_{send}^{s^{\prime }},q_{send}/2^{l}\right\}.$$  □

#### 4.2.2. ProVerif

ProVerif [40,41] is a powerful and appropriate tool for analyzing and verifying protocol security. We use it to verify our protocol’s security. (1)Some functions and queries are also defined, as shown in Figure 6a,b.(2)Figure 6c shows the defined events and queries. Among them, we define eight queries. The first three queries prove the session key’s security, while the other five queries prove the protocol’s correctness. In addition, we also defined eight events. Event UserStarted() indicates that $U_{i}$ begins authentication, event UserAuthed() indicates that $U_{i}$ successfully authenticated, event ControlServerAcUser() represents CS authenticating $U_{i}$ successfully, event ControlServerAcCloudServer() represents CS authenticating $S_{j}$ successfully, event CloudServerAcControlServer() indicates that $S_{j}$ successfully authenticates CS, event UserAcControlServer() represents $U_{i}$ authenticating CS successfully, event UserAcCloudServer() represents $U_{i}$ authenticating $S_{j}$ successfully, and event CloudServerAcUser() represents CS authenticating $U_{i}$ successfully.(3)Figure 7a–c shows $U_{i}$’s, $S_{j}$’s, and CS’s processes, respectively. Finally, Figure 8 presents the results. The first three results demonstrate that attackers cannot obtain SK, and the last five outcomes demonstrate that the protocol is correct and reasonable. Therefore, our protocol can successfully pass the verification of ProVerif and prevent common attacks.

### 4.3. Informal Security Analysis

In this subsection, an informal analysis is adopted to demonstrate the common security requirements of the proposed protocol.

#### 4.3.1. Man-in-the-Middle Attacks

A computes SK by intercepting messages on the open channel. Let us suppose that message $M_{1}$ is intercepted and A attempts to calculate $SK=h(r_{i}⊕HPW_{i}\|r_{j}\|r_{k}\|SID_{j})$ but they cannot obtain the values of $ID_{CS},HPW_{i},SID_{j}$. Therefore, A cannot use the message $\left\{B_{1},B_{2},B_{4},B_{7}\right\}$ on the open channel to calculate $r_{i},r_{j},r_{k},B_{3},B_{5}$; change any values; or successfully pass the authentication of CS; thus, they cannot successfully calculate SK. Consequently, the proposed protocol can guard against MITM attacks.

#### 4.3.2. Insider Attacks

Case one: Assume that a malicious attacker A obtains $\left\{QID_{j},n_{j},TID_{i},HPW_{i}\right\}$ stored in the CS database. They use the message on the open channel to compute $r_{i}=B_{1}⊕h\left(ID_{CS}\|HPW_{i}⊕SID_{j}\right)$. However, A cannot obtain the values of $ID_{CS},SID_{j}$, and thus, $r_{i}$ cannot be calculated. Similarly, because A cannot obtain the values of $A_{3},SID_{j}$, A cannot calculate $r_{j}=B_{4}⊕h\left(A_{3}\|SID_{j}\right)$ and $r_{k}=h\left(HPW_{i}\|r_{i}\right)⊕B_{10}$. Therefore, the session key cannot be computed using A. Therefore, our protocol can prevent insider attacks.

Case two: Assume that the attacker A is an insider of the cloud server and obtains the information $A_{3}^{*},QID_{j}$ stored in it. They then try to intercept the information on the open channel and to calculate the session key $SK=h(r_{i}⊕HPW_{i}\|r_{j}\|r_{k}\|SID_{j})$. They intercepted $B_{4}$ and tried to calculate $r_{j}=B_{4}⊕h\left(A_{3}\|SID_{j}\right)$ but cannot calculate the value of $A_{3}$ and thus cannot obtain the value of $r_{j}$. Similarly, A attempts to intercept $B_{6}$ and $B_{7}$ to calculate $\left(r_{i}⊕HPW_{i}\right)=B_{6}⊕A_{3}$, and $r_{k}=h(A_{3}\|r_{j}\left\|SID_{j}\right)⊕B_{7}$. However, they cannot obtain the value of $A_{3}$ and thus cannot calculate $\left(r_{i}⊕HPW_{i}\right)$ and $r_{k}$, so they cannot successfully calculate SK.

By analyzing these two situations, we can prove that our protocol can resist insider attacks.

#### 4.3.3. DDoS Attacks

During the login and authentication phase, $U_{i}$ sends service request message $M_{1}=\left\{TID_{i},A_{1},B_{1},B_{2},B_{3},TS_{1}\right\}$ to $S_{j}$. After $S_{j}$ receives $M_{1}$, whether the timestamp is valid is checked first. If the timestamp is valid, $S_{j}$ performs the following calculation. Therefore, if attacker A wants to launch DDoS attacks, it must be within a valid time, and it is not possible in this protocol to deny a service only by sending a huge service request. Therefore, the protocol is immune to this attack.

#### 4.3.4. Masquerading Attacks

Case one: Attacker A attempts to impersonate any legitimate user, cloud server, or control server. Suppose that A obtains the information $\left\{QID_{j},n_{j},TID_{i},HPW_{i}\right\}$ stored in CS and intercepts the messages $\left\{M_{1},M_{2},M_{3},M_{4}\right\}$ on the public channel. A wants to impersonate a legitimate $U_{i}$ by calculating $B_{3}=h(TID_{i}\|ID_{CS}\left\|n_{i}⊕x\right)⊕HPW_{i}$, but A cannot obtain values of $ID_{CS}$ and $\left(n_{i}⊕x\right)$. Therefore, they cannot successfully calculate the value of $B_{3}$ and cannot impersonate a legitimate user by changing $B_{3}$ to pass the verification of CS and thus cannot pretend to be a legitimate user.

Case two: Similarly, A wants to impersonate a $S_{j}$ through $B_{5}=h(r_{j}\|A_{3}\left\|SID_{j}\right)$ but cannot obtain values of $r_{j},A_{3}$ and $SID_{j}$, so they cannot pass the verification of CS. Therefore, A cannot successfully impersonate a legal $S_{j}$. It can be concluded that the proposed protocol can resist impersonation attacks.

To sum up, our protocol can resist masquerading attacks.

#### 4.3.5. Identity Theft Attacks

Suppose that an attacker A obtains the user’s smart card and tries to impersonate a legitimate user to establish a session with the cloud server and the control server. They obtain $\left\{A_{1},A_{2},ID_{CS},Gen\left(\cdot \right),Rep\left(\cdot \right),\tau_{i}\right\}$ and try to calculate the authentication value $B_{3}=h(TID_{i}\|ID_{CS}\left\|n_{i}⊕x\right)⊕HPW_{i}$ by intercepting the information on the open channel. Because they cannot obtain $PW_{i},\tau_{i}$, they cannot calculate $HPW_{i}=h\left(PW_{i}\|\tau_{i}\right)$ and thus cannot successfully calculate $B_{3}$ and pass the authentication of CS. Therefore, our protocol can resist identity theft attacks.

#### 4.3.6. Replay Attacks

According to our defined attacker model, an attacker A can forward the intercepted message to the receiver on the open channel and prove that they are a legitimate entity if the receiver authenticates the message. However, each transmitted message has a timestamp. If A transmits a previously intercepted message, the recipient rejects the request because of the invalid timestamp. Thus, the protocol is resistant to replay attacks.

#### 4.3.7. Perfect Forward Secrecy

In our protocol, $SK=h(r_{i}⊕HPW_{i}\|r_{j}\|r_{k}\|SID_{j})$. Case one: Suppose that an attacker A can obtain *x* but $SID_{j}$ and $HPW_{i}$ cannot be computed and A cannot obtain random numbers $r_{i}$, $r_{j}$, and $r_{k}$. Therefore, there is no way to calculate the current SK or the previous SK, so the proposed protocol can provide perfect forward security.

Case two: Assume that an attacker *A* obtains the user’s password $PW_{i}$ to attack. Because the user’s biological information $B_{i}$ cannot be obtained, A cannot compute $HPW_{i}$, where $HPW_{i}=h\left(PW_{i}\|\tau_{i}\right)$. Additionally, $\left\{r_{i},r_{j},r_{k},SID_{j}\right\}$ is unknown. A cannot successfully compute SK.

Case three: Assume that an attacker A can obtain the private value $A_{3}^{*}$ of a cloud server for an attack. Because the identity $SID_{j}$ of the $S_{j}$ cannot be obtained, A cannot calculate $A_{3}$, where $A_{3}=h\left(SID_{j}\|x⊕n_{j}\right)$. Furthermore, A cannot calculate $r_{j}$ and $r_{k}$; here, $r_{i}=B_{1}⊕h\left(ID_{CS}\|HPW_{i}⊕SID_{j}\right)$ and $r_{k}=h(A_{3}\|r_{j}\left\|SID_{j}\right)⊕B_{7}$. Additionally, $HPW_{i}$ is unknown, and A cannot successfully calculate SK.

Therefore, the proposed protocol can provide perfect forward security.

#### 4.3.8. Session Key Disclosure Attacks

It is assumed that the attacker A attempts to intercept the transmission of information on the open channel. Even if the attacker intercepts the messages $M_{1}-M_{5}$, they cannot compute $r_{j}=B_{4}⊕h\left(A_{3}\|SID_{j}\right)$, $r_{k}=h(A_{3}\|r_{j}\left\|SID_{j}\right)⊕B_{7}$, and $\left(r_{i}⊕HPW_{i}\right)=B_{6}⊕A_{3}$ because they cannot obtain the values of $HPW_{i},SID_{j},A_{3}$. Obviously, they cannot compute the session key $SK=h(r_{i}⊕HPW_{i}\|r_{j}\|r_{k}\|SID_{j})$ by intercepting the information on the public channel. Therefore, our proposed protocol can resist session key disclosure attacks.

#### 4.3.9. Mutual Authentication

In the login and authentication phase, CS verifies the user and cloud server through $B_{3}$ and $B_{5}$, respectively, and $B_{8}$ and $B_{11}$ are the values of $U_{i}$ and $S_{j}$ used to verify mutual identity, respectively. Although $B_{3}$ and $B_{5}$ are transmitted over the open channel, the values of $\left(n_{i}⊕x\right)$, $HPW_{i}$, $r_{j}$, and $A_{3}$ cannot be obtained by an attacker A. Similarly, $B_{8}$ and $B_{11}$ are also transmitted over the open channel, but A cannot obtain the values of $r_{k}$ and SK, and thus, the protocol cannot break by changing the authentication value. Hence, our protocol can provide mutual authentication.

#### 4.3.10. Privacy and Anonymity

An attacker A attempts to identify a user by intercepting messages on the open channel. However, in our proposed method, A can only obtain $U_{i}$’s pseudo identity $TID_{i}$. Thus, A cannot compute the user’s real $ID_{i}$. Similarly, A can only obtain the $S_{j}$’s pseudo identity $QID_{j}$. A cannot determine the true identity of $U_{i}$ and $S_{j}$ based on the pseudo identity, which protects the privacy of $U_{i}$ and $S_{j}$. Therefore, our protocol can provide privacy and anonymity.

#### 4.3.11. Traceability and Non-Repudiation

When cloud server finds that $U_{i}$ has bad behavior, it will report to CS, and CS will find the value of the user’s $HPW_{i}$ according to $TID_{i}$, which can be used to identify $U_{i}$. Therefore, once a user exhibits malicious behavior, CS can track the user, which ensures traceability. Since the transmitted message $M_{1}=\left\{TID_{i},A_{1},B_{1},B_{2},B_{3},TS_{1}\right\}$ contains the value of authenticating the user’s identity $B_{3}$, once a legitimate user exhibits bad behavior, CS will verify the user’s identity according to $B_{3}=h(TID_{i}\|ID_{CS}\left\|n_{i}⊕x\right)⊕HPW_{i}$. If the verification is passed, this indicates that the bad behavior is indeed sent by the user, and the user cannot deny it. Therefore, non-repudiation is guaranteed.

#### 4.3.12. Integrity

Integrity is the guarantee that an attacker A cannot change the transmitted information. Even if A is able to successfully tamper with the information, the system will detect and discover that the information has been modified.

It is assumed that an attacker A can intercept and tamper with the messages $\{M_{1},M_{2}$, $M_{3},M_{4}\}$ transmitted on the open channel. For example, A intercepts and tampers with message $M_{1}$, where $M_{1}=\left\{TID_{i},A_{1},B_{1},B_{2},B_{3},TS_{1}\right\}$. If A tampers with $TID_{i}$, CS cannot retrieve $HPW_{i}$ and the authentication is suspended. If A tampers with $A_{1},B_{1},B_{2},B_{3}$, then CS calculates that $B_{3}^{\prime }$ is not equal to the received $B_{3}=h(TID_{i}\|ID_{CS}\left\|n_{i}⊕x\right)⊕HPW_{i}$, which indicates that the user is not legal or $M_{1}$ is tampered with, and the authentication is suspended. Similarly, if an attacker intercepts and tampers with $\left\{M_{2},M_{3},M_{4}\right\}$, all three entities will be checked accordingly. Therefore, the proposed protocol can ensure the integrity of information.

#### 4.3.13. Confidentiality

From the Section 4.3.2 (Insider Attacks) and Section 4.3.4 (Masquerading Attacks), it can be seen that the attacker cannot obtain $SK=h(r_{i}⊕HPW_{i}\|r_{j}\|r_{k}\|SID_{j})$. Therefore, it can be seen that our protocol ensures confidentiality.

## 5. Security and Performance Comparison

We compared the protocols of Amin et al. [13], Martinez et al. [17], Zhou et al. [18], and Kang et al. [19] in terms of performance and security. The specific comparison results are described in the following subsection.

### 5.1. Security Comparison

This subsection compares the five protocols in terms of security. Specifically, indicates that the security characteristics are met, and × indicates that they are not met. In addition, S1–S8 are defined as follows: S1: Perfect forward secrecy; S2: Man-in-the-middle attack; S3: Mutual authentication; S4: Impersonation attack; S5: Replay attack; S6: Temporary value disclosure attack; S7: Offline password-guessing attack; and S8: Insider attack.

Table 3 lists the security results. From Table 3, Zhou et al.’s [18] scheme was unable to provide perfect forward security and cannot prevent replay, user and server impersonation, and temporary value disclosure attacks. The protocol of Kang et al. [19] cannot resist offline password-guessing attacks; Amin et al.’s [13] protocol cannot prevent insider or impersonation attacks; and the protocol of Martinez et al. [17] was unable to prevent impersonation or replay attacks, or even enable mutual authentication. The proposed protocol can evidently prevent many common attacks.

### 5.2. Performance Comparison

We calculated the time required by the user and server. To estimate the user’s computing cost, we developed an app that uses the Java pairing library, signature library, and symmetric encryption/decryption function to calculate the running time of various operations. We used smart phones produced by different manufacturers to imitate the user. We ran various operations on the following mobile phones ten times and used the average value as the reference time. Table 4 lists the results of various operations on different mobile phones. D1 is a Huawei Mate 30 mobile phone with a harmony operating system, Huawei Kirin 990 processor, and 8G running memory; D2 is a Redmi Note 9 Pro mobile phone with an Android operating system, Qualcomm Snapdragon 750 g CPU, and 8G of RAM; and D3 represents a Oneplus phone with an Android operating system, Snapdragon 865 processor, and 8G running memory. Table 5 and Figure 9 present the comparative results of the client calculation costs of the five protocols. Table 5 shows that the protocol of Amin et al. [13] requires $9T_{h}$, the protocol of Martinez et al. [17] requires $11T_{h}+T_{de}$, Zhou et al.’s [18] protocol requires $10T_{h}$, Kang et al.’s protocol [19] requires $8T_{h}$, and our proposed protocol requires $T_{m}+10T_{h}$. Although we are not as good as Amin et al. [13], Zhou et al. [18], and Kang et al. [19] in terms of performance, we are better than them in terms of security.

We used a computer with a windows 10 operating system, Intel (R) core (TM) i5-8500CPU@3.00GHz3.00G processor, and 8 GB memory, to simulate the server’s computational costs. IntelliJ idea software version 2019.3 was used for development. It is based on the Java pairing library, signature library, and symmetric encryption/decryption function. Various operations were run 10 times on this computer, and the average value was used as the reference time. Table 4 lists the results. Table 6 shows the comparison results. As shown in Table 6, the time required for our proposed protocol was only four hashes more than Kang et al.’s [19] and Amin et al.’s [13] protocols, that is, 0.0208 ms.

To calculate the communication cost, we set the length of one-way hash *H* as 256 bits, timestamp *T* to 32 bits, string *s* to 160 bits, identity ID to 160 bits, random number $Z_{p}^{*}$ to 160 bits, and encryption operation *E* to 256 bits. In addition, assume that the fuzzy extractor needs to store 8 bits. From this definition, the protocol of Zhou et al. [18] can be concluded to require 3|ID|+10|s|+6|H|+3|T|, that of Amin et al. [13] requires 3|ID|+8|s|+6|H|+3|T|, that of Martinez et al. [17] requires $3|ID|+18|s|+|H|+2|E|+2|Z_{p}^{*}|$, that of Kang et al. [19] requires 3|ID|+10|s|+6|H|+3|T|, and our protocol requires 3|ID|+15|s|+5|H|+3|T|. Table 7 and Figure 10 present the detailed comparison results. Evidently, our protocol has a lower communication cost than that of Martinez et al. [17], although the communication cost is higher than that of the other three protocols. Our protocol also provides higher security; their protocols have been proven to have various security problems. Therefore, our proposed protocol is secure, has a relatively good performance, and is suitable for cloud computing environments.

For the storage costs, we consider the parameters required to store each entity in each entity in the registration phase. The number of numbers required for various parameters is the same as discussed above. The comparison results are shown in Figure 11. It can be seen from the figure that our storage cost is slightly higher than the protocols of Amin et al. [13] and Kang et al. [19].

In terms of energy costs, due to the strong computing power and good performance of the server, we do not consider its energy cost. We use “ampere” APP to measure the current and voltage of each mobile phone the three devices during operation, and the results are shown in the Table 8. According to the formula W=U·I·t, the power consumption required by each device was calculated. The results are shown in Table 9 and Figure 12. The energy costs are different for different devices. It can be seen from the figure that, although our protocol is not the best, it is better than that of Martinez et al. [17] and our protocol is secure.

## 6. Conclusions and Disscussion

IoT-enabled cloud computing environment is an open environment, and its main threat is data leakage. Because a large number of customers’ privacy data are stored on the cloud server, once the data is leaked, it will lead to not only the disclosure of trade secrets, intellectual property rights, and personal privacy but also a devastating blow to the enterprise. In addition, the communication information of various entities is transmitted on the open channel. Attackers can launch attacks by intercepting the information on the open channel. Moreover, from Table 1, we can know that most of the existing schemes have been attacked, such as man in the middle attack, simulation attack, etc.

In this paper, we propose a protocol to solve the security problem in this environment. To verify the security, an informal security analysis was conducted, and the ProVerif and ROR models were adopted for formal security analysis. Finally, the protocol’s security and performance were measured against those of other protocols. The proposed protocol can be concluded to satisfy basic security requirements. Therefore, our protocol is more suitable for this environment.

## Figures and Tables

**Figure 1 sensors-22-03858-f001:**
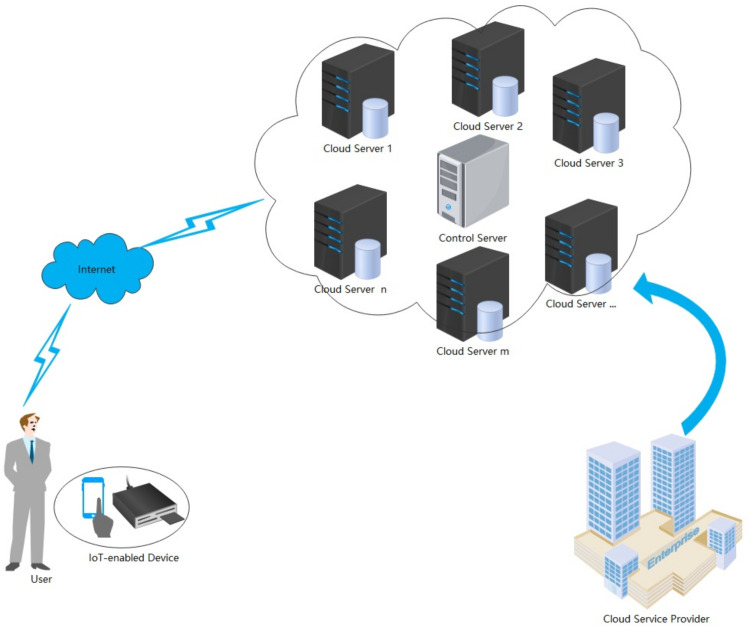
Architecture of IoT-enabled cloud computing.

**Figure 2 sensors-22-03858-f002:**
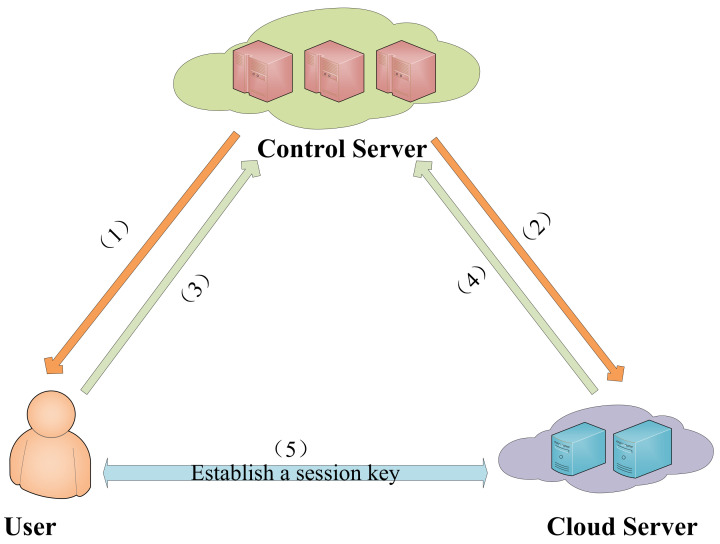
Information exchange process.

**Figure 3 sensors-22-03858-f003:**
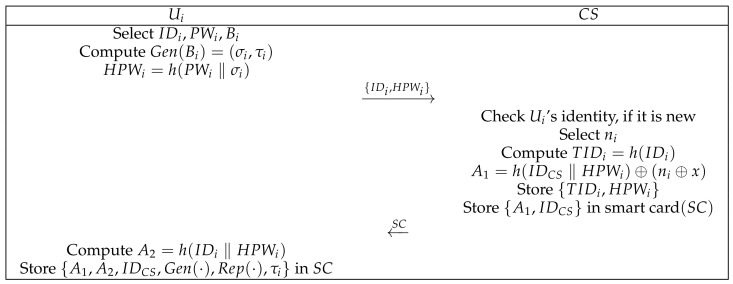
User registration phase.

**Figure 4 sensors-22-03858-f004:**
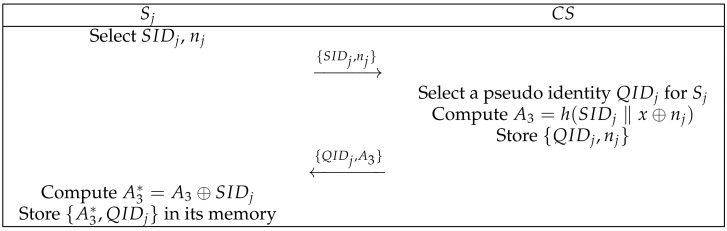
Cloud server registration phase.

**Figure 5 sensors-22-03858-f005:**
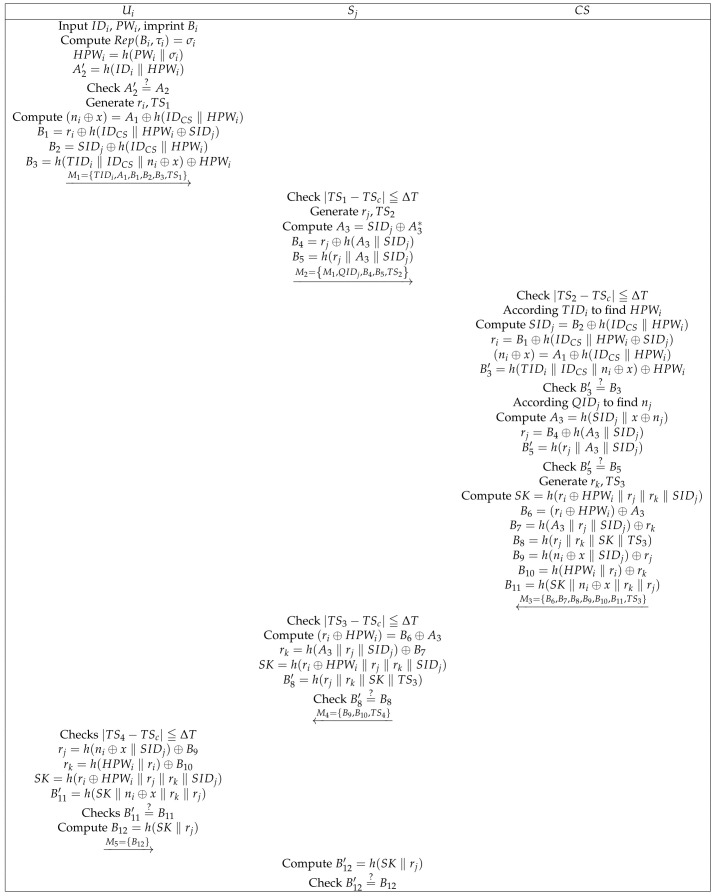
Login and authentication phase.

**Figure 6 sensors-22-03858-f006:**
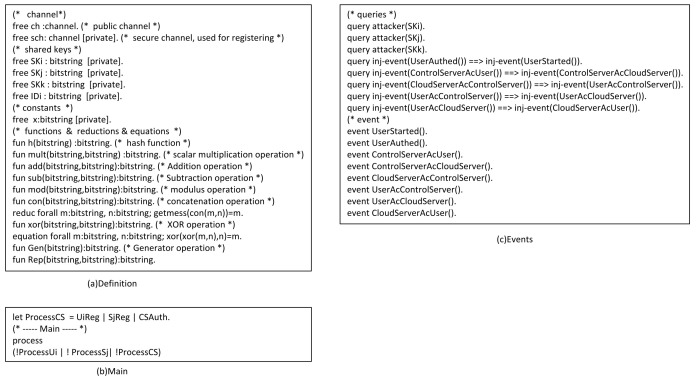
Definitions.

**Figure 7 sensors-22-03858-f007:**
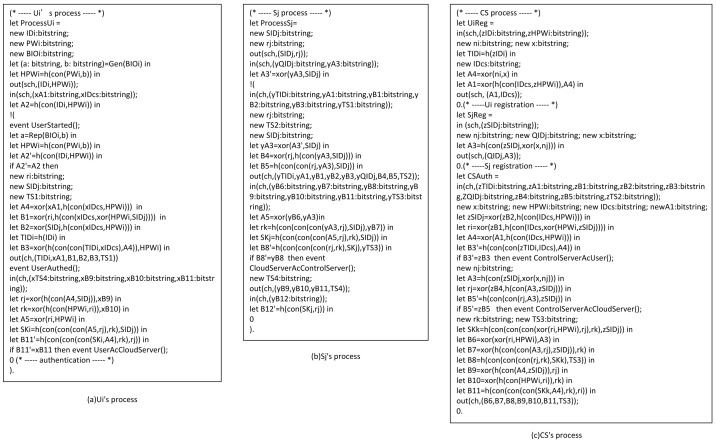
Process.

**Figure 8 sensors-22-03858-f008:**
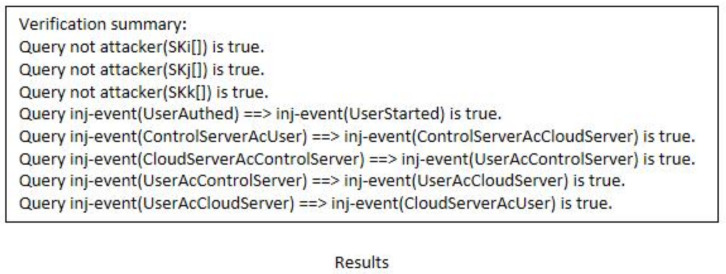
Results.

**Figure 9 sensors-22-03858-f009:**
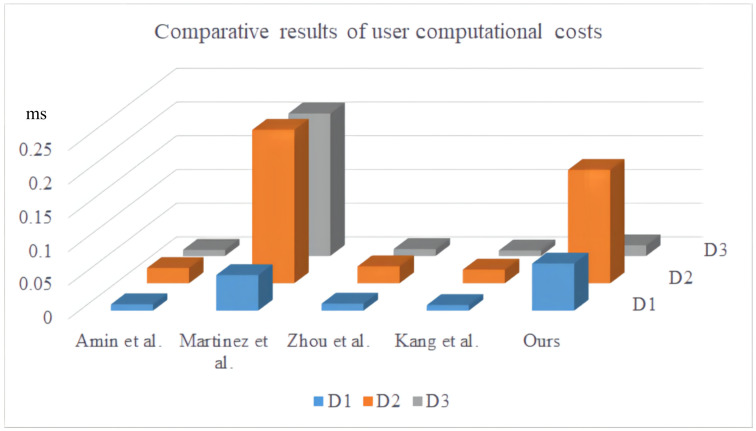
Comparative results of user computational costs [13,17,18,19].

**Figure 10 sensors-22-03858-f010:**
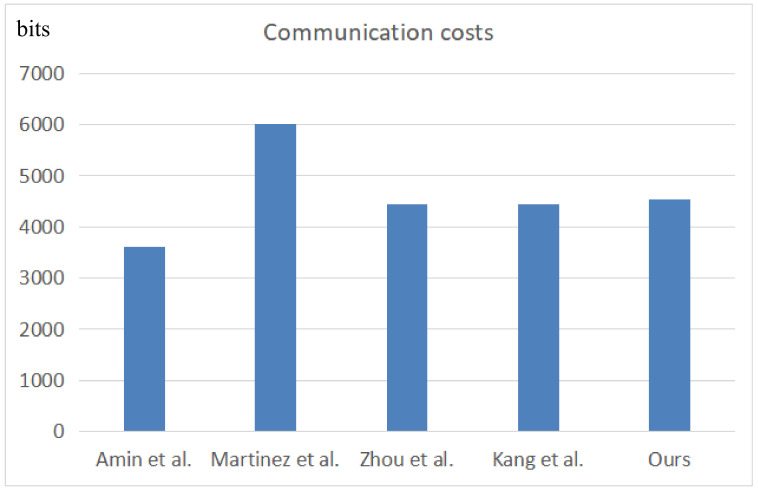
Comparative results of communication costs [13,17,18,19].

**Figure 11 sensors-22-03858-f011:**
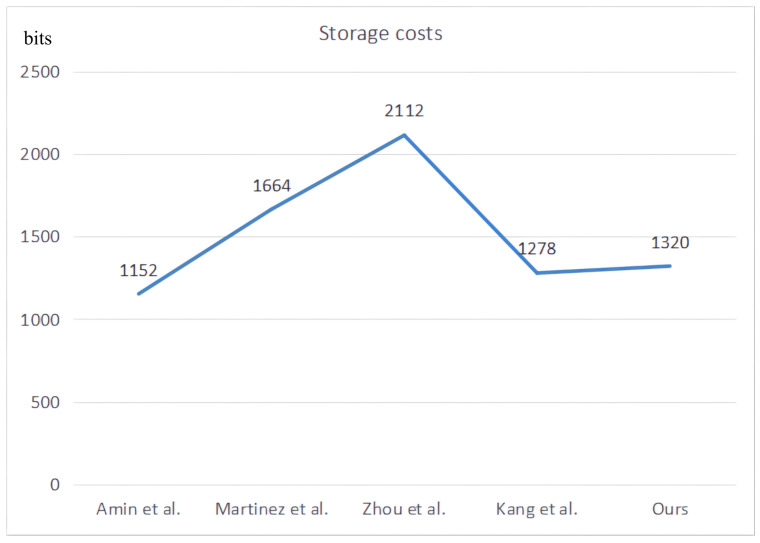
Comparative results of storage costs [13,17,18,19].

**Figure 12 sensors-22-03858-f012:**
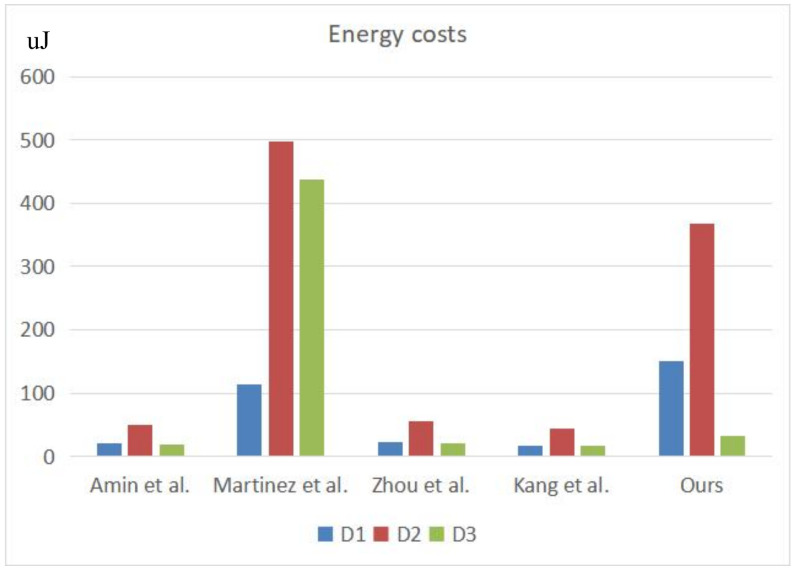
Comparative results of energy costs [13,17,18,19].

**Table 1 sensors-22-03858-t001:** A summary of authentication protocols.

Protocols	Advantages	Shortcomings
Turkanovic et al. [22]	(1) Provides user anonymity (2) Can resist offline password- guessing attacks	(1) Cannot resist insider (2) Cannot resist user impersonation attacks
Wazid et al. [23]	(1) Can resist user impersonation attacks (2) Provides user anonymity (3) Provides perfect forward security	-
Wu et al. [24]	(1) Can resist temporary value (2) Can resist offline passowrd- guessing attacks	(1) Cannot resist sensor capture attacks (2) Cannot resist denial of service attacks (3) Cannot provide perfect forward security
Tsai and Lo [25]	(1) Can resist temporary value disclosure attacks (2) Provides perfect forward security	(1) Cannot resist server impersonation attacks
Irshad et al. [26]	(1) Can resist user impersonation attacks Provides Perfect forward security	(1) Lacks user registration and revocation phases
Amin et al. [13]	(1) Can resist temporary value disclosure attacks (2) Can resist insider attacks	(1) Cannot prevent insider attacks (2) Cannot resist impersonation attacks
Martinez et al. [17]	(1) Can resist user impersonation attacks (2) Can resist offline password- guessing attacks (3) Provides user anonymity	(1) Cannot prevent impersonation attacks (2) Cannot resist session key exposure attacks (3) Cannot achieve mutual authentication
Zhou et al. [18]	(1) Provides user anonymity (2) Can achieve mutual authentication (3) Can resist insider attacks	(1) Cannot prevent replay attacks (2) Cannot prevent impersonation attacks (3) Cannot prevent temporary value disclosure attacks (4) cannot provide perfect forward security
Kang et al. [19]	(1) Can resist impersonation attacks (2) Can achieve mutual authentication	(1) Cannot resist offline password-guessing attacks

**Table 2 sensors-22-03858-t002:** Notations.

Notations	Meanings
$S_{j}$	The *j*th cloud server
$SID_{j}$	The $S_{j}$’s identity
$U_{i}$	The *i*th user
$ID_{i}$	$U_{i}$’s identity
$PW_{i}$	$U_{i}$’s password
$B_{i}$	$U_{i}$’s biological information
$HPW_{i}$	$U_{i}$’s pseudo password
SC	Smart card
CS	Control server
$ID_{CS}$	CS’s identity
*x*	The secret key of CS
$TID_{i}$	$U_{i}$’s pseudo identity
$QID_{j}$	$S_{j}$’s pseudo identity
h(·)	Hash function
Gen(·),Rep(·)	Fuzzy extraction function
$\tau_{i},\sigma_{i}$	Two parameters generated by the fuzzy extractor [34], where $\tau_{i}$ is public and $\sigma_{i}$ is private.
$TS_{1},TS_{2},TS_{3},TS_{4}$	Timestamps

**Table 3 sensors-22-03858-t003:** Comparisons of security.

Security Properties	[13]	[17]	[18]	[19]	Ours
S1	✓	✓	×	✓	✓
S2	×	✓	✓	✓	✓
S3	✓	×	✓	✓	✓
S4	×	×	×	✓	✓
S5	✓	×	×	✓	✓
S6	✓	✓	×	✓	✓
S7	✓	✓	✓	×	✓
S8	×	✓	✓	✓	✓

**Table 4 sensors-22-03858-t004:** The computational costs of complex operations.

Operations	Symbolic	D1 (ms)	D2 (ms)	D3 (ms)	Server (Cloud, Contorl)
Symmetric Decryption	$T_{de}$	0.04125	0.2	0.2	0.1347
Symmetric Encryption	$T_{en}$	0.2	0.0392	0.0591	4.7
Hash function	$T_{h}$	0.00103	0.00251	0.00102	0.0052
Fuzzy function	$T_{f}$	0.05665	0.143	0.00561	-

**Table 5 sensors-22-03858-t005:** Comparative results of user computational costs.

Protocols	User	D1 (ms)	D2 (ms)	D3 (ms)
Amin et al. [13]	$9T_{h}$	0.0093	0.0226	0.0092
Martinez et al. [17]	$11T_{h}+T_{de}$	0.0526	0.2275	0.2112
Zhou et al. [18]	$10T_{h}$	0.0103	0.0251	0.0102
Kang et al. [19]	$8T_{h}$	0.0082	0.0201	0.0082
Ours	$T_{f}+10T_{h}$	0.0697	0.1681	0.0158

**Table 6 sensors-22-03858-t006:** Comparative results of server computational costs.

Protocols	Cloud Server	Control Server	Total (ms)
Amin et al. [13]	$4T_{h}$	$10T_{h}$	0.0728
Martinez et al. [17]	$6T_{h}+2T_{de}+T_{en}$	$34T_{h}+2T_{en}$	14.5774
Zhou et al. [18]	$7T_{h}$	$20T_{h}$	0.1404
Kang et al. [19]	$3T_{h}$	$11T_{h}$	0.0728
Ours	$5T_{h}$	$13T_{h}$	0.0936

**Table 7 sensors-22-03858-t007:** Comparisons in terms of communication and storage costs.

Protocols	Number of Rounds	Communication Costs (Bits)	Storage Costs (Bits)	Security
Amin et al. [13]	5	3680	1152	Insecure
Martinez et al. [17]	6	6016	1664	Insecure
Zhou et al. [18]	4	4448	2112	Insecure
Kang et al. [19]	2	4000	1278	Cannot resist offline password guessing attack
Ours	5	4544	1320	Provable secruity

**Table 8 sensors-22-03858-t008:** Voltage and current of devices.

Devices	U (V)	I (mA)
D1	4.08	531
D2	610	3.58
D3	508	4.08

**Table 9 sensors-22-03858-t009:** Energy costs.

Protocols	D1 (uJ)	D2 (uJ)	D3 (uJ)
Amin et al. [13]	20.148	49.354	19.068
Martinez et al. [17]	113.957	496.814	437.74
Zhou et al. [18]	22.315	54.813	21.14
Kang et al. [19]	17.751	43.894	16.996
Ours	151.004	367.097	32.748

## Data Availability

The data are contained within the article.

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
