# Peer review of "Rotating behind Security: A Lightweight Authentication Protocol Based on IoT-Enabled Cloud Computing Environments"

_sensors, 2022, doi:10.3390/s22103858_

Round 1
Reviewer 1 Report
The authors proposed a lightweight authentication protocol designed for IoT-enabled cloud computing environments. However, I have the following concerns:
1. (Line 12) The Introduction section is very small. The motivation of the work should be highlighted in this section. The motivation of the paper is not clear. Why is your proposal needed? What are the challenges involved? What solutions already exist for the problem you want to solve? What are their limitations and drawbacks?
2. (Line 58) In the related work section, the authors only point out the problems of different papers and do not summarize the common problems. It is not clear from the related work whether the authors' contributions are unresolved problems in the current research. For instance, there is no information on the drawbacks and limitations of these works. Do the existing works already solve the problem? Why is your proposal needed?
3. (Figure 1. Architecture of IoT-enabled cloud computing) The network model of the proposed system is not clear. The exchange of messages is not clear. The author's description of the functions of each entity is not sufficiently detailed and organized. The authors also do not provide a description of the level of credibility of each entity.
4. What are the threats against IoT-enabled cloud computing ? What are the threats of such kind of communication environment? The discussion in the paper is incomplete. Important assumptions like CK adversary model are not considered. What is the role of different communicating parties? Who are semi-trusted, who are trusted, who are not trusted? This information is not clear in the paper.
5. If an IoT device is instead considered vulnerable as normal, the scheme becomes vulnerable to a stolen smartcard in presence of a privileged insider attack.
6- (Line 234) The authors should add an informal security analysis for the following security requirements: Confidentiality, Integrity, privacy and anonymity, Traceability and non-repudiation, Perfect Forward Secrecy (PFS), Resistance against replay attacks, Resistance against DDoS attacks, Resistance against session key disclosure attack, Resistance against masquerading attacks, Resistance against identity theft attacks, Resistance against traffic analysis attacks.
7- (Line 264)The author should add the performance metrics for storage cost, and energy cost.
Reviewer 2 Report
Few comments and suggestions:
In Section 3, for the user registration phase, what is B_i?
What are the fuzzy extraction functions Gen() and Rep()? How are they implemented?
For the computation cost in Table 3, which elliptic curves have been used to implement map-to-point, bilinear pairing and point multiplication operations?
Elliptic curve computations are known to be significantly more expensive than symmetric encryption, hash functions, etc. In Table 3, how is point multiplication so cheap? how is it even orders of magnitude faster than hash functions and symmetric encryption / decryption when it is expected to be the opposite? Please verify the implementation results for performance / compute cost. This changes the entire user compute cost / performance comparison with previous work. Unlike Table 3, the server compute cost for point multiplication in Table 5 seems to be closer to expected.
In Section 5, it is mentioned that the proposed protocol involves one point multiplication. Which step in the protocol requires this elliptic curve operation?
Table 5 provides server computation costs. Figure 8 provides compute cost for cloud server and control server. How are separate costs for cloud and control servers in Figure 8 obtained from the data in Table 5?
Figure 7 should use separate column charts for D1, D2, D3 rather than stacked chart with all three together.
Figure 8 should use separate column charts for cloud server and control server rather than stacked chart with both together.
Reviewer 3 Report
* The paper proposes a secure authentication protocol for IoT devices. The protocol is validated formally with (ProVerif) and informally. According to the obtained results, it is more secure than other existing protocols. Furthermore, the performance was measured with several smartphones and a self-developed application, as well as with a computer; the performance is good, which is more suitable for the use of IoT devices.
* Acronyms are not always well defined at their first use:
- QR (line 45), ROR (141, although defined in the final acronyms table)
* Some minor typing mistakes or suggestions:
- line 18: "amounts data" -> "amounts of data"
- line 34: "devices, The" -> "devices. The"
- line 87: "that the" -> "that it" / "that they" ??
- line 90: "This section introduces our protocol. It includes" -> "This section introduces our protocol, which includes"
- line 90-91: "three phases: ..." -> "three phases: (1) user registration, (2) cloud server registration, and (3) login and authentication.
- line 91: "The subsections" -> "The following subsections"
- I think that the word "sever" has a typo, as it should be "server" (for "Control Server" or "Cloud server"), but it appears in many points: line 95, 101, 102, caption of Figure 3, 221, 222, 224, 225, 226, Fig. 5(b) (the same 5), 331.
- line 109: "inprints" -> "imprints"
- line 157: "n" -> "z"
- line 168: "they" -> "it" ??
- line 170: "Send(Y,M)" -> "Send(Z,M)"
- line 179: "execute(z)" -> "execute(Z)" (uppercase)
- line 201: "This that" ??
- line 215: "operations and queries" -> "operations" ??
- line 246: "Suppose" -> "Let's suppose"
- line 292: "Table 2" -> "Table 4" ??
* To improve the readability:
- Subsections 3.1, 3.2 and 3.3 shown nearly the same as figures 2,3,4. Perhaps, the phases could be described in a more textual form, or at least, to include an introduction text before (1),(2),(3), etc.
- Table 1 (notations) does not include: B_i, HPW_i, Delta_i, Tau_i, SIDj, TS_1, TS_SC. The meaning of ID_i could be "Ui's identity".
- line 145: "of A" -> "of an attacker A" (as below, but this is the first appearance of A)
- In Subsection 4.1.2 ProVerif, from my point of view the items (1)... are not necessary. It could be written only as a normal text. For example, (3),(4),(5),(6) could be written: "Fig.6 (a),(b),(c) show U_i's, S_j's and CS's processes, respectively. Finally, ... Fig.5(d) ..."
* About tables/figures:
- Figure 1 could be shown on pg.2 instead of pg.3, as it is referenced from pg.1.
- In section 5, a figure and its associated table with data are referenced together, (e.g. Table 4 and Fig. 7, or Table 6 and Fig. 8), but both are separated. Perhaps, it could be better to put both together. Perhaps, Table 3 and and Table 5 could be put together.
* About references/bibliography:
- The bibliography seems to be complete.
- Some references show the year twice: 4, 15, 19, 21, 34, 35
- "things" -> "Things" in: 8, 9
- In caption of Figure 9 and Table 7, "communicational cost" -> "communication cost" (and in line 332, also "Table 3" -> "Table 7")?
Round 2
Reviewer 1 Report
the authors have appropriately addressed the comments and the revisions made are satisfactory.
Author Response
Response to Reviewer #1 (2nd Round Revision)
Reviewer 1: The authors have appropriately addressed the comments and the revisions made are satisfactory.
Author response: We thank for Reviewer’s careful reviewing and comments.
Reviewer 2 Report
Feedback from previous round of review have been addressed.
One minor comment - Tables 5, 6, 7 compare the computation, communication and storage costs with previous work. The authors have rightly noted that the proposed protocol is not as efficient as many of the previous work but the loss in efficiency is due to increase in security. It will be nice to include the security aspect in one of these tables, e.g., Table 7, to indicate the additional security features offered by the proposed protocol.
Author Response
Response to Reviewer #2 (2nd Round Revision)
Reviewer 2: One minor comment - Tables 5, 6, 7 compare the computation, communication and storage costs with previous work. The authors have rightly noted that the proposed protocol is not as efficient as many of the previous work but the loss in efficiency is due to increase in security. It will be nice to include the security aspect in one of these tables, e.g., Table 7, to indicate the additional security features offered by the proposed protocol.
Author response: We thank for Reviewer’s careful reviewing and comments. We have added the security aspect in Table 7 to indicate the additional security features offered by the proposed protocol.
Author action: In Page 16, Table 7 has been modified.
Reviewer 3 Report
The authors have fixed the typos and improved the manuscript, according to the recommendations and comments from the reviewers.
In Figure 6. Definitions (current version), the word "Sever" should be "Server" (several times).
Author Response
Response to Reviewer #3 (2nd Round Revision)
Reviewer 3: In Figure 6. Definitions (current version), the word "Sever" should be "Server" (several times).
Author response: We thank for Reviewer’s careful reviewing and comments. We have corrected these typing mistakes in Figure 6.
Author action: In page 11, Figure 6 has been modified.